# Vascular Endothelial Growth Factor Receptor 2 (VEGFR2) rs2071559 Gene Polymorphism and the Risk of Gliomas: A Systematic Review and Meta-Analysis

**DOI:** 10.3390/jcm13154332

**Published:** 2024-07-25

**Authors:** Patricia Diana Prasetiyo, Eka Julianta Wahjoepramono

**Affiliations:** 1Department of Pathology, Faculty of Medicine, Pelita Harapan University, Tangerang 15811, Indonesia; 2Department of Neurosurgery, Faculty of Medicine, Pelita Harapan University, Tangerang 15811, Indonesia; ekabrain@gmail.com

**Keywords:** gene, glioma, neoplasm, polymorphism, VEGF

## Abstract

Background: A glioma is a form of tumor that is abundant in blood vessels. Vascular endothelial growth factor receptor (VEGFR) and its receptor 2 (VEGFR2) are important in the process of angiogenesis. The relationship between VEGFR2 rs2071559 and glioma development is currently uncertain. The present study aims to analyze the correlation between VEGFR2 rs2071559 gene polymorphism and the susceptibility to gliomas. Methods: A thorough search was carried out in the Cochrane Library, Scopus, and Medline databases from inception until 20 February 2024 using a mix of pertinent keywords. We used random-effects models to examine the odds ratio (OR) and reported the results together with their respective 95% confidence intervals (CIs). Results: A total of six studies were incorporated. The results of our meta-analysis indicated that all genetic models of VEGFR2 rs2071559 gene polymorphism, starting from dominant (OR 1.40; *p* < 0.00001), recessive (OR 1.52; *p* < 0.0001), CC genotype (OR 1.78; *p* < 0.00001), CT genotype (OR 1.30; *p* < 0.0001), and C allele (OR 1.41; *p* < 0.00001), were associated with a higher risk of developing gliomas. The subgroup analysis revealed a higher OR for studies with a sample size of ≥500, originated from Asia, with a mean age of ≥42.3 years, and a male prevalence of <57%. Conclusions: This study suggests that VEGFR2 rs2071559 gene polymorphism is associated with a higher risk of gliomas.

## 1. Introduction

A glioma is a neoplasm that arises from neuroepithelial tissue. It is the most prevalent malignant neoplasm in the central nervous system. The incidence rate of this type of tumor represents almost 80% of all malignant tumors that occur intracranially. A glioma is a very aggressive tumor that exhibits rapid growth and has a brief period of disease progression [1,2]. Glioma tumor cells have a high propensity for infiltrating the adjacent healthy brain tissue and the interface between them, rendering total surgical excision challenging. In addition, the likelihood of relapse following surgery is rather high, and patients typically experience a short survival period. The tumor’s uncontrolled growth of cells, decreased cell death, ability to invade surrounding tissues, and formation of new blood vessels have contributed to its reputation as one of the most difficult tumors to treat in the field of neurosurgery [3,4]. Based on recent statistics, the 2-year survival rates for patients with low-grade astrocytoma, degenerative astrocytoma, and glioblastoma multiforme, even with active treatment such as surgery, radiotherapy, or pharmacological chemotherapy, are only 66%, 45%, and 9%, respectively [5]. The primary cause of this situation stems from a lack of thorough comprehension regarding the characteristics of gliomas, encompassing their pathophysiology, molecular biological alterations, individual variances, diagnostic and prognostic markers, and specific treatment strategies [6,7]. Consequently, researching the etiology of gliomas becomes a pivotal aspect in addressing this issue.

A glioma is a type of tumor that occurs in the central nervous system and is characterized by an abundance of blood vessels. Vascular endothelial growth factor (VEGF) and its receptor (VEGFR2) play significant roles in the development and regulation of blood vessels and the process of angiogenesis. VEGFR2 exhibits robust ligand-induced tyrosine phosphorylation in intact cells, serving as the primary mediator of vascular endothelial cell proliferation and differentiation. Additionally, this receptor plays a role in facilitating the aggressive growth of gliomas [8,9]. A study conducted by Liu G et al. [10] revealed a substantial increase in the expression level of VEGFR2 in glioma patients compared to healthy controls. This finding suggests that the abnormal expression of this receptor may have an impact on nerve cell activity, potentially leading to the development of gliomas [10]. Currently, the understanding of the association between VEGFR2, specifically rs2071559, and the susceptibility to gliomas remains uncertain. The objective of our work is to conduct a thorough analysis of the correlation between VEGFR rs2071559 and the likelihood of developing gliomas.

## 2. Materials and Methods

### 2.1. Eligibility Criteria

This review was compiled based on the guidelines from the PRISMA statement [11]. In this review, we only included case–control studies that involved populations of patients diagnosed with a glioma (as the case group) and healthy patients without any history of gliomas (as the control group). Those studies must have data on the kinase insert domain receptor (KDR) or VEGFR2 gene polymorphism, specifically rs2071559, which may be in the form of major (T) and minor (C) alleles, as well as the distribution of genotype variations (TT, CT, and CC) in both groups of patients.

Meanwhile, research that meets any of the following exclusion criteria will be excluded from this review: (1) Research conducted in a pediatric population; (2) Research conducted on brain malignancies other than glioma; (3) Insufficient data on rs2071559; (4) Not accessible in its full-text form; (5) Research conducted using methods other than a case–control design.

### 2.2. Search Strategy and Study Selection

A thorough examination of the available literature was conducted, with a specific emphasis on publications written in the English language. The search spanned from database inception to 20 February 2024 and was conducted in three significant international databases: Medline, Scopus, and the Cochrane Library. The search terms employed for the literature review were as follows: “(kinase insert domain receptor OR KDR OR vascular endothelial growth factor receptor 2 OR VEGFR2 OR rs2071559) AND (glioma OR glial tumor OR glial cell tumors OR glioblastoma OR astrocytoma OR oligodendroglioma OR ependymoma)”. The preliminary search was carried out by two authors (P.D.P. and E.J.W.). The same two authors (P.D.P. and E.J.W.) additionally verified the citations exported to the reference manager to ensure they were consistent and complete. In order to discover any other pertinent publications, the process of citation tracking was employed. This involved scrutinizing the references of the discovered studies, tracing their citations, and investigating the linked articles. Furthermore, a subsequent investigation of gray literature sources was carried out. The same two authors (P.D.P. and E.J.W.) separately assessed the titles and abstracts of potentially relevant articles, excluding items that were not related to this study. The process of determining eligibility based on the full text of the studies was carried out by the same two authors, and any differences in judgment were resolved through discussion.

### 2.3. Data Extraction

Two authors (P.D.P. and E.J.W.) independently retrieved essential data from the eligible articles including the characteristics of the participants (i.e., age, sex distribution, glioma histological type, and tumor grading) in addition to the study characteristics (i.e., author last name, year of publication, country, and number of participants).

### 2.4. Risk of Bias Assessment

Two independent reviewers (P.D.P. and E.J.W.) conducted an examination of the potential bias in each article using standardized assessment tools. The Newcastle–Ottawa Scale (NOS) was utilized to evaluate the overall quality of every observational study. This scale encompasses assessments about the recruitment of study participants, the comparability of different groups, and the outcomes of the studies. The instrument has a score range from 0 to 9. In the research context, studies that achieved a total score of 7 or more were considered to have an acceptable level of quality (good quality) [12].

### 2.5. Statistical Analysis

For outcomes in the form of dichotomous variables, we calculated an odds ratio (OR) with a 95% confidence interval (95% CI) using the Mantel–Haenszel technique. This was done to compare the VEGFR2 (rs2071559) gene polymorphism between the two groups of patients. In this review, we opted to use random-effects models due to the anticipation of substantial heterogeneity resulting from variations in population characteristics. We used three prevalent genetic models (dominant, recessive, and allele type) and three additional genotype models (CC vs. TT, CT vs. TT, and CC vs. CT) to thoroughly evaluate the association between the VEGFR2 (rs2071559) gene polymorphism and the susceptibility to gliomas. In this study, we employed the I-squared (I^2^) statistic to evaluate the variation among studies. We classified I^2^ values of ≤25%, 26–50%, and >50% as indicating low, moderate, and high heterogeneity, respectively [13]. A subgroup analysis was performed on each model based on the sample size, region, mean age, and sex distribution. A publication bias analysis was conducted when there were at least 10 studies available for each outcome of interest. The statistical analyses were conducted using Review Manager 5.4, an application developed by the Cochrane Collaboration (United Kingdom).

## 3. Results

### 3.1. Study Selection and Characteristics

We conducted a search across three databases: Medline (*n* = 362), Cochrane Library (*n* = 254), and Scopus (*n* = 220). A total of 836 citations were found and reviewed. Then, 817 citations were excluded as duplicates and deemed ineligible after title/abstract screening. Out of 19 records reviewed, 13 papers were discarded for the following reasons: six lacked data on the outcomes of interest, four did not contain data related to rs2071559, and three were review articles. Ultimately, the final analysis contained six publications [14,15,16,17,18,19] comprising 2347 glioma patients and 2503 healthy controls (Figure 1). Of the six pieces of research, five originated from China, while the remaining one came from Brazil. The sample size in the glioma group ranged from 157 to 766 among the examined studies, while in the healthy control group, it varied from 160 to 824. The predominant histological types of gliomas in the investigations were astrocytoma and glioblastoma. Table 1 provides a detailed summary of the histological type and tumor grading for each study included in this analysis.

### 3.2. Quality of Study Assessment

All the case–control studies in our analysis were of high quality according to the Newcastle–Ottawa Scale (NOS), scoring between 8 and 9 (Table 2). All the studies met the criteria for inclusion in the meta-analysis.

### 3.3. Classical Model

#### 3.3.1. Dominant (CC + CT vs. TT)

Our pooled analysis of six studies (*n* = 4828) showed that the dominant model of KDR/VEGFR2 (rs2071559) gene polymorphism (CC + CT vs. TT) was associated with a higher risk of developing gliomas [OR 1.40 (95% CI 1.24–1.57), *p* < 0.001, I^2^ = 0%, random-effects models] (Figure 2A).

A subgroup analysis revealed variations in the OR for this model (Table 3). Based on the sample size, studies with a sample size < 500 (OR = 1.44, *p* = 0.02) showed a higher OR than studies with a sample size ≥ 500 (OR = 1.22, *p* < 0.001). Based on the region, only studies from countries in Asia showed a significant result (*p* < 0.001), while studies from countries outside of Asia showed a non-significant result (*p* = 0.31). Based on the mean age of participants, studies with a mean age of ≥42.3 years (OR = 1.49, *p* < 0.001) showed a higher OR than studies with a mean age of <42.3 years (OR = 1.37 *p* < 0.001). Based on the sex distribution, studies with a male sex prevalence of <57% (OR = 1.54, *p* < 0.001) showed a higher OR than studies with a male sex prevalence of ≥57% (OR = 1.34, *p* < 0.001).

#### 3.3.2. Recessive (CC vs. CT + TT)

Our pooled analysis of six studies (*n* = 4828) showed that the recessive model of KDR/VEGFR2 (rs2071559) gene polymorphism (CC vs. CT + TT) was associated with a higher risk of developing gliomas [OR 1.52 (95% CI 1.25–1.85), *p* < 0.001, I^2^ = 24%, random-effects models] (Figure 2B).

A subgroup analysis revealed variations in the OR for this model (Table 3). Based on the sample size, only studies with a sample size ≥ 500 showed a significant result (*p* < 0.001), while studies with a sample size < 500 showed a non-significant result (*p* = 0.23). Based on the region, only studies from countries in Asia showed a significant result (*p* < 0.001), while studies from countries outside of Asia showed a non-significant result (*p* = 0.58). Based on the mean age of participants, studies with a mean age of ≥42.3 years (OR 1.60, *p* = 0.02) showed a higher OR than studies with a mean age of <42.3 years (OR = 1.48 *p* < 0.001). Based on the sex distribution, studies with a male sex prevalence of <57% (OR = 1.74, *p* < 0.001) showed a higher OR than studies with a male sex prevalence of ≥57% (OR = 1.46, *p* = 0.007).

#### 3.3.3. Allele (C vs. T)

Our meta-analysis from a total of five studies (*n* = 6010) showed that the C allele of KDR/VEGFR (rs2071559) gene polymorphism was associated with a higher risk of developing gliomas than the T allele [OR 1.41 (95% CI 1.27–1.57), *p* < 0.001, I^2^ = 0%, random-effects models] (Figure 2C).

A subgroup analysis revealed variations in the OR for this model (Table 3). Based on the sample size, only studies with a sample size ≥ 500 showed a significant result (*p* < 0.001), while studies with a sample size < 500 showed a non-significant result (*p* = 0.08). Based on the region, only studies from countries in Asia showed a significant result (*p* < 0.001), while studies from countries outside of Asia showed a non-significant result (*p* = 0.33). Based on the mean age of participants, studies with a mean age of <42.3 years (OR 1.45, *p* < 0.001) showed a higher OR than studies with a mean age of ≥42.3 years (OR = 1.38 *p* < 0.001). Based on the sex distribution, studies with a male sex prevalence of <57% (OR = 1.45, *p* < 0.001) showed a higher OR than studies with a male sex prevalence of ≥57% (OR = 1.38, *p* < 0.001).

### 3.4. Additional Model

#### 3.4.1. CC vs. TT

Our meta-analysis from a total of six studies (*n* = 2697) showed that the genetic model consisting of CC genotypes of KDR/VEGFR2 (rs2071559) gene polymorphism was associated with a higher risk of gliomas than the model consisting of TT genotypes [OR 1.78 (95% CI 1.42–2.24), *p* < 0.001, I^2^ = 32%, random-effects models] (Figure 3A).

A subgroup analysis for this genotype model based on the sample size, region, mean age, and sex distribution can be seen in Table 3.

#### 3.4.2. CT vs. TT

Our meta-analysis from a total of six studies (*n* = 4156) showed that the genetic model consisting of CT genotypes of KDR/VEGFR2 (rs2071559) gene polymorphism was associated with a higher risk of gliomas than the model consisting of TT genotypes [OR 1.30 (95% CI 1.15–1.47), *p* < 0.001, I^2^ = 0%, random-effects models] (Figure 3B).

A subgroup analysis for this genotype model based on the sample size, region, mean age, and sex distribution can be seen in Table 3.

#### 3.4.3. CC vs. CT

Our meta-analysis from a total of six studies (*n* = 2803) showed that the genetic model consisting of CC genotypes of KDR/VEGFR2 (rs2071559) gene polymorphism was associated with a higher risk of gliomas than the model consisting of CT genotypes [OR 1.33 (95% CI 1.11–1.59), *p* = 0.002, I^2^ = 0%, random-effects models] (Figure 3C).

A subgroup analysis for this genotype model based on the sample size, region, mean age, and sex distribution can be seen in Table 3.

### 3.5. Publication Bias

A funnel plot analysis was used to evaluate publication bias. The current study did not examine publication bias because there were less than 10 studies included for each outcome of interest. Consequently, the evaluation of publication bias lacks the same level of robustness as when there are more than 10 studies available for analysis [20,21].

## 4. Discussion

Our systematic review and meta-analysis indicates that all genetic models of KDR/VEGFR2 (rs2071559) gene polymorphism significantly increase the risk of gliomas. These models include the dominant model (CT + CT vs. TT), recessive model (CC vs. CT + TT), CC genotype, CT genotype, and C allele. Our subgroup analysis revealed a significant association between the KDR/VEGFR2 (rs2071559) gene polymorphism and studies with a sample size of ≥500 from Asia. Studies with a sample size of <500 from outside of Asia did not show any significant results. The subgroup analysis also revealed a greater relationship between the KDR/VEGFR2 (rs2071559) gene polymorphism and glioma occurrence in studies where the mean age was ≥42.3 years and the male sex prevalence was <57%, as shown by a higher odds ratio.

Our meta-analysis results validate the significance of genetic variables, particularly SNPs, in the development of gliomas. A glioma is a vascular-rich kind of tumor. Angiogenesis in malignant gliomas is driven by many pro-angiogenic cytokines, with VEGF being the predominant signaling molecule. This cytokine family consists of six isoforms of VEGF, namely VEGF-A, VEGF-B, VEGF-C, VEGF-D, VEGF-E, and placental growth factor [22]. VEGF-A is widely recognized as the primary mediator in the promotion of tumor growth caused by a lack of oxygen (hypoxia). VEGF signaling is facilitated by receptor tyrosine kinases such as VEGFR-1, VEGFR-2, and VEGFR-3. This signaling pathway is responsible for various tasks, including promoting the growth of new blood vessels (pro-angiogenic activity), increasing the permeability of blood vessels, and stimulating the movement of endothelial cells [23]. VEGF has been demonstrated to have a synergistic effect with other growth factors, and the combined effects of VEGF with other factors surpass the effects of each component alone in promoting angiogenesis. The interaction between VEGF and its receptors on the endothelial cell membrane triggers the activation of endothelial cells, leading to the secretion of matrix metalloproteinase (MMP) into the surrounding tissue. These MMP enzymes are responsible for breaking down the extracellular matrix (ECM), which is necessary for the proliferation and migration of endothelial cells. Furthermore, the coadministration of VEGF-A with fibroblast growth factor-2 (FGF-2) or platelet-derived growth factor-BB (PDGF-BB) demonstrated a robust synergistic impact on promoting angiogenesis both in laboratory settings and in living organisms [24,25].

VEGF is crucial for the survival and growth of gliomas. The expression of VEGF mRNA was detected in low-grade gliomas and showed increased levels in high-grade gliomas [24]. The development of gliomas is initiated by the activation of VEGFR-1 mRNA in endothelial cells, while the transition to malignancy is facilitated by the combined activity of the VEGFR-1/VEGFR-2 genes [26]. Elevated levels of VEGF mRNA expression were detected in the necrotic areas of glioblastoma tumors, thereby stimulating vascular proliferation and advancing the course of human glioblastoma. The study found that the increased expression of VEGF and VEGF-R1 in low-grade astrocytomas was strongly linked to a poor prognosis, similar to high-grade lesions. This suggests that the levels of VEGF and VEGFR can be used as prognostic biomarkers and provide valuable information for deciding the treatment approach [24,27].

The single nucleotide polymorphism (SNP) examined in our meta-analysis was rs2071559. Rs2071559 is situated in the area of the VEGFR2 promoter that interacts with nuclear proteins, likely transcription factors. The rs2071559 (- 604T > C) mutation reduces the ability of the VEGFR2 promoter to bind to specific transcription factors, resulting in a lower amount of VEGFR2 and its soluble form (sVEGFR2). We hypothesize that the inhibitory impact of sVEGFR2 on tumor angiogenesis is more potent than the promoting influence of VEGFR2. When the expression of both forms declines simultaneously, the inhibitory impact of sVEGFR2 is less potent than the promoting impact of VEGFR2. Both factors work together to enhance tumor growth and the development of new blood vessels, reflecting the progression of gliomas [16,19].

This systematic review and meta-analysis is the first to thoroughly examine the correlation between KDR/VEGFR2 (rs2071559) gene polymorphism and the risk of gliomas. Our study contributes to the extensive body of evidence on the involvement of KDR/VEGFR2 rs2071559 in the development of several diseases. The prior meta-analysis conducted by Seyedmirzaei H et al. [28] exclusively examined the levels of VEGF in patients with glioma. They found that VEGF levels were significantly elevated in glioma patients compared to healthy controls. However, the genetic aspect, particularly the involvement of the VEGFR2 gene polymorphism in glioma formation, was not addressed in their analysis nor their discussion. Genetic factors have a crucial role in the development of diseases, including cancers like gliomas. Hence, our present meta-analysis diverges significantly from the prior study conducted by Seyedmirzaei H et al. [28]. In contrast to their focus on VEGF levels, we explore the genetic aspect, specifically exploring the impact of gene polymorphism in the VEGF receptor, namely VEGFR2, on glioma development. In the context of CNS disease, a recent study by Bruzaite A et al. [29] discovered a correlation between KDR/VEGFR2 rs2071559 and the development of pituitary adenomas. Zhang W et al. [30] conducted a study that revealed an association between KDR/VEGFR2 rs2071559 and the development and recurrence of stroke. Regarding cancer, KDR/VEGFR2 rs2071559 has been identified as a predictive factor in breast cancer [31], colorectal cancer (CRC) [32], hepatocellular carcinoma (HCC) [33], non-small cell lung cancer (NSCLC) [34], and renal cell carcinoma (RCC) [35].

This investigation has limitations. First, the majority of the study’s research was exclusively carried out in Asian countries, specifically China. Therefore, the applicability of these results may be limited, especially when considering groups that are not of Asian origin. Second, most of the research analyzed involved a limited number of participants, with only two studies including over 1000 participants. This small sample size from the included studies may introduce bias into our analysis; therefore, our results must be interpreted with caution. Third, our current research solely investigates the impact of a single SNP, specifically KDR/VEGFR2 rs2071559, on the development of gliomas. Our current study did not assess the other SNPs from the KDR/VEGFR2 gene as well as other genes that may contribute to the development of gliomas. In addition, it is important to bear in mind that environmental and lifestyle factors might also exert an influence on the development of gliomas. Future research should explore the role of KDR/VEGFR2 gene polymorphism in countries outside of Asia and include a larger sample size. We also still encourage future research that explores the role of other SNPs from KDR/VEGFR2 gene polymorphism in the development of gliomas to provide a better understanding about how changes in VEGFR2 function or expression may contribute to the development of gliomas. Issues related to the therapeutic implications (e.g., drug sensitivity, treatment efficacy) of the VEGFR2 rs2071559 gene polymorphism can also be explored in future studies.

## 5. Conclusions

The findings of our systematic review and meta-analysis indicate an association between KDR/VEGFR2 (rs2071559) gene polymorphism and the risk of developing gliomas. An elevated risk of gliomas was observed in all the genetic models, including dominant (CC + CT vs. TT), recessive (CC vs. CT + TT), CC genotype, CT genotype, and C allele of KDR/VEGFR2 (rs2071559) gene polymorphism. Further subgroup analysis revealed that this association is particularly stronger in Asia-originated studies with a sample size ≥ 500 that have a mean age of ≥42.3 years and a male sex prevalence of <57%. Therefore, our study unveils evidence regarding the involvement of the KDR/VEGFR2 rs2071559 gene polymorphism in the development of gliomas, especially among Asian populations. Further research is needed to explore the potential involvement of other SNPs in the VEGFR2 gene or neighboring genes in the development of gliomas, as our current investigation focused on only one SNP (rs2071559). Subsequent research could investigate this matter.

## Figures and Tables

**Figure 1 jcm-13-04332-f001:**
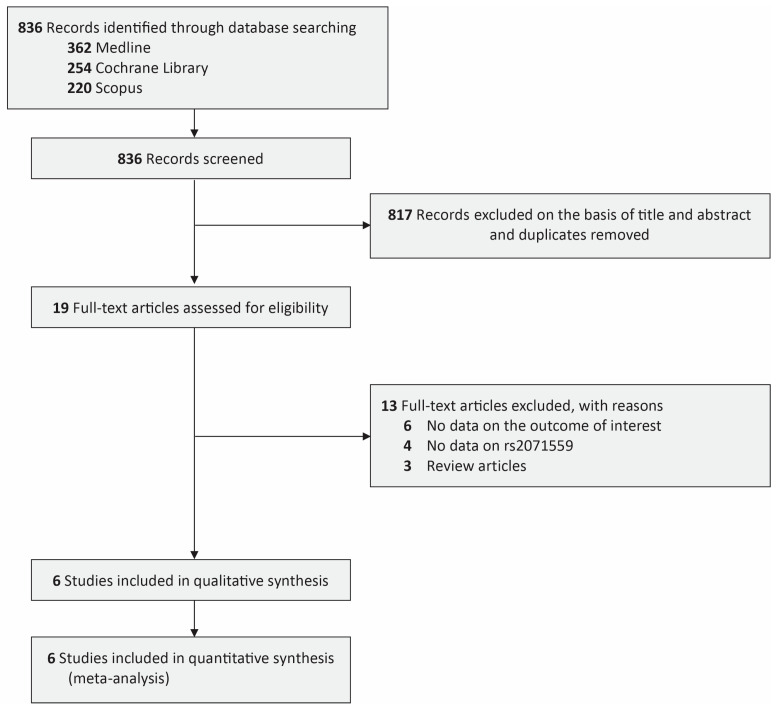
PRISMA diagram of the detailed process of selection of studies for inclusion in the systematic review and meta-analysis.

**Figure 2 jcm-13-04332-f002:**
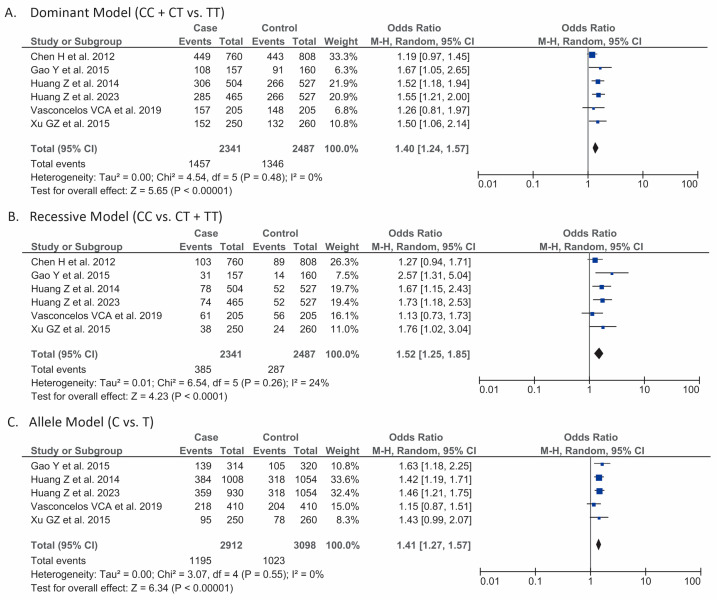
Forest plot that demonstrates the association between classical genetic models of kinase insert domain protein (KDR)/vascular endothelial growth factor receptor 2 (VEGFR2) rs2071559 gene polymorphism: dominant (CC + CT vs. TT) (**A**) [14,15,16,17,18,19], recessive (CC vs. CT + TT) (**B**) [14,15,16,17,18,19], and allele (C vs. T) (**C**) [15,16,17,18,19] and the risk of gliomas.

**Figure 3 jcm-13-04332-f003:**
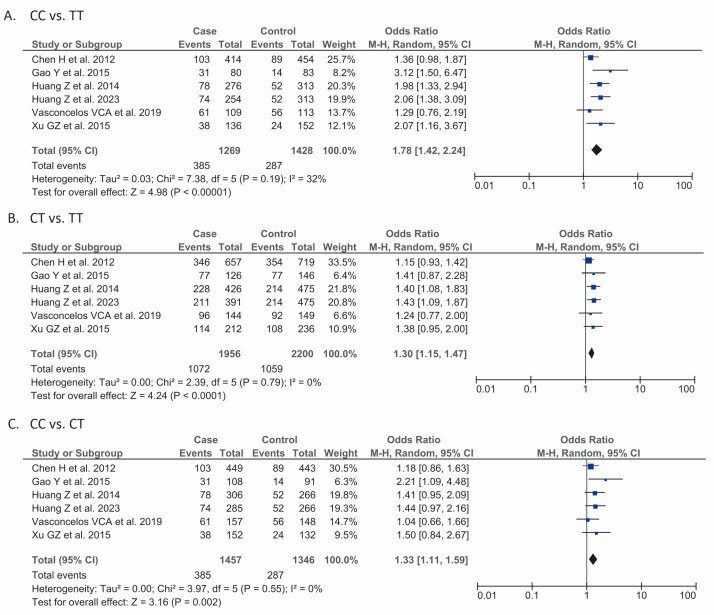
Forest plot that demonstrates the association between additional genetic models of kinase insert domain protein (KDR)/vascular endothelial growth factor receptor 2 (VEGFR2) rs2071559 gene polymorphism: CC vs. TT (**A**) [14,15,16,17,18,19], CT vs. TT (**B**) [14,15,16,17,18,19], and CC vs. CT (**C**) [14,15,16,17,18,19] and the risk of gliomas.

**Table 1 jcm-13-04332-t001:** Characteristics of included studies.

Study ID		HWE Test	Cases					Control		
Authors	Country		Sample Size	Age (Mean ± SD)	Male (%)	Histological Type	Tumor Grade	Sample Size	Age (Mean ± SD)	Male (%)
Chen H et al. [14] 2012	China	0.880	766	42.2 ± 16	59.1%	Astrocytoma = 37% Glioblastoma = 31.5% Other glioma = 31.5%	Not reported	824	41.5 ± 18.4	59.5%
Gao Y et al. [15] 2015	China	0.247	157	42.7 ± 10.2	61.8%	Astrocytoma = 100%	1–2 = 62.4% 3–4 = 37.6%	160	41.3 ± 12.9	59.4%
Huang Z et al. [16] 2014	China	0.451	504	42.3 ± 15.8	59.3%	Astrocytoma = 34.3% Glioblastoma = 31.5% Other glioma = 34.1%	1 = 9.7% 2 = 31.9% 3 = 18.1% 4 = 40.2%	527	40.2 ± 16.3	55.4%
Huang Z et al. [17] 2023	China	0.424	465	42.2 ± 15.4	58.5%	Astrocytoma = 37.2% Glioblastoma = 34.2% Oligodendroglioma = 10.1% Ependymoma = 14.4% Mixed glioma = 4.1%	1 = 10.5% 2 = 34.6% 3 = 19.6% 4 = 35.3%	527	40.2 ± 16.3	55.4%
Vasconcelos VCA et al. [18] 2019	Brazil	0.390	205	51 ± 15.3	65.8%	High grade glioma = 100% (type not specified)	3 = 17.5% 4 = 81.9%	205	48 ± 8.1	48.7%
Xu GZ et al. [19] 2015	China	>0.05	250	40.4 ± 11.8	54%	Type not specified	Not reported	260	40.6 ± 11.4	54.2%

HWE = Hardy–Weinberg equilibrium, SD = standard deviation.

**Table 2 jcm-13-04332-t002:** Newcastle–Ottawa quality assessment of observational studies.

First Author, Year	Study Design	Selection ^a^	Comparability ^b^	Outcome ^c^	Total Score	Result
Chen H et al. [14] 2012	Case–control	****	**	***	9	Good
Gao Y et al. [15] 2015	Case–control	****	**	***	9	Good
Huang Z et al. [16] 2014	Case–control	***	**	***	8	Good
Huang Z et al. [17] 2023	Case–control	****	**	***	9	Good
Vasconcelos VCA et al. [18] 2019	Case–control	****	**	***	9	Good
Xu GZ et al. [19] 2015	Case–control	****	**	***	9	Good

^a^ (1) Is the case definition adequate? (2) Representativeness of the cases; (3) Selection of controls; (4) Definition of controls. ^b^ (1) Comparability of cases and controls on the basis of design or analysis (maximum two stars). ^c^ (1) Ascertainment of exposure; (2) Same method of ascertainment for cases and controls; (3) Non-response rate. ** = scores 2 point; *** = scores 3 point; **** = scores 4 point.

**Table 3 jcm-13-04332-t003:** Summary of meta-analysis results for different models of KDR/VEGFR2 rs2071559 polymorphism.

Outcome	Intervention	Included Studies	Outcome (95% CI)	*p*-Value	I^2^ (%)
**Dominant model (CC + CT vs. TT)**					
Sample size	<500	2	OR: 1.44 (1.05–1.99)	0.02	0
	≥500	4	OR: 1.22 (1.22–1.61)	<0.001	20
Region	Asia	5	OR: 1.41 (1.24–1.60)	<0.001	7
	Outside of Asia	1	OR: 1.26 (0.81–1.97)	0.31	-
Age	<42.3	3	OR: 1.37 (1.14–1.65)	<0.001	35
	≥42.3	3	OR: 1.49 (1.22–1.81)	<0.001	0
Male sex prevalence	<57%	2	OR: 1.54 (1.25–1.89)	<0.001	0
	≥57%	4	OR: 1.34 (1.15–1.56)	<0.001	8
**Recessive model (CC vs. CT + TT)**					
Sample size	<500	3	OR: 1.63 (0.73–3.63)	0.23	75
	≥500	4	OR: 1.52 (1.26–1.84)	<0.001	0
Region	Asia	5	OR: 1.60 (1.32–1.94)	<0.001	11
	Outside of Asia	1	OR: 1.13 (0.73–1.73)	0.58	-
Age	<42.3	3	OR: 1.48 (1.19–1.85)	<0.001	3
	≥42.3	3	OR: 1.60 (1.06–2.40)	0.02	55
Male sex prevalence	<57%	2	OR: 1.74 (1.27–2.37)	<0.001	0
	≥57%	4	OR: 1.46 (1.11–1.93)	0.007	44
**Homozygote model (CC vs. TT)**					
Sample size	<500	2	OR: 1.93 (0.82–4.57)	0.13	73
	≥500	4	OR: 1.76 (1.41–2.20)	<0.001	19
Region	Asia	5	OR: 1.88 (1.47–2.40)	<0.001	33
	Outside of Asia	1	OR: 1.29 (0.76–2.19)	0.34	-
Age	<42.3	3	OR: 1.72 (1.27–2.32)	<0.001	37
	≥42.3	3	OR: 1.90 (1.24–2.91)	0.003	48
Male sex prevalence	<57%	2	OR: 2.06 (1.49–2.87)	<0.001	0
	≥57%	4	OR: 1.69 (1.22–2.34)	0.002	49
**Heterozygote model (CT vs. TT)**					
Sample size	<500	2	OR: 1.32 (0.94–1.86)	0.11	0
	≥500	4	OR: 1.30 (1.14–1.48)	<0.001	0
Region	Asia	5	OR: 1.31 (1.15–1.49)	<0.001	0
	Outside of Asia	1	OR: 1.24 (0.77–2.00)	0.38	-
Age	<42.3	3	OR: 1.27 (1.09–1.48)	0.002	0
	≥42.3	3	OR: 1.37 (1.11–1.69)	0.003	0
Male sex prevalence	<57%	2	OR: 1.41 (1.14–1.76)	0.002	0
	≥57%	4	OR: 1.26 (1.08–1.46)	0.003	0
**Heterozygote model (CC vs. CT)**					
Sample size	<500	2	OR: 1.45 (0.70–3.01)	0.32	67
	≥500	4	OR: 1.33 (1.09–1.63)	0.004	0
Region	Asia	5	OR: 1.39 (1.14–1.68)	<0.001	0
	Outside of Asia	1	OR: 1.04 (0.66–1.66)	0.86	-
Age	<42.3	3	OR: 1.31 (1.04–1.65)	0.02	0
	≥42.3	3	OR: 1.38 (0.97–1.98)	0.08	36
Male sex prevalence	<57%	2	OR: 1.46 (1.05–2.03)	0.02	0
	≥57%	4	OR: 1.29 (1.02–1.63)	0.03	15
**Allele model (C vs. T)**					
Sample size	<500	2	OR: 1.35 (0.96–1.90)	0.08	62
	≥500	3	OR: 1.44 (1.27–1.63)	<0.001	0
Region	Asia	4	OR: 1.46 (1.30–1.64)	<0.001	0
	Outside of Asia	1	OR: 1.15 (0.87–1.51)	0.33	-
Age	<42.3	2	OR: 1.45 (1.23–1.71)	<0.001	0
	≥42.3	3	OR: 1.38 (1.16–1.64)	<0.001	30
Male sex prevalence	<57%	2	OR: 1.45 (1.23–1.71)	<0.001	0
	≥57%	3	OR: 1.38 (1.16–1.64)	<0.001	30

## Data Availability

Data included in article/referenced in article.

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
