# Peer review of "Vascular Endothelial Growth Factor Receptor 2 (VEGFR2) rs2071559 Gene Polymorphism and the Risk of Gliomas: A Systematic Review and Meta-Analysis"

_jcm, 2024, doi:10.3390/jcm13154332_

Round 1
Reviewer 1 Report
Comments and Suggestions for Authors
The abstract must be structured.
VEGFR - is an abbreviation for Vascular Endothelial Growth Factor Receptor, correct this aspect.
Enter the keywords in alphabetical order
Introduction:
References are placed in square brackets before the point and at the end of the quoted paragraph, not after each statement.
Line 40, please adapt the diagnoses according to the last classification (the same aspect in the discussion chapter).
Line 46: add studies: https://doi.org/10.1038/s41582-019-0220-2 to support reference 6.
Paragraph 48-63 - can be excluded, it has nothing to do with the subject itself
Lines 69-72 – the statements are repetitive.
Materials and methods
Given the fact that the study has only two authors, they are probably the same ones who performed the searches and systematized the analysis of the cases. Therefore, reformulate the paragraphs (the idea of ​​several uncredited authors in the article is induced).
The company and state must be added to the software used in the analysis.
Results
Table 1 - degrees are written in Arabic numerals (review this aspect throughout the text). Add lines to the table, where there are many histological types it is difficult to follow the column they belong to.
For the p value, 3 digits are passed after the point
Table 3 – find another term for Country (eg. Region?)
Discussions:
Start discussions with the other unspecified role of VEGF – angiogenesis (https://doi.org/10.1007/s12035-020-01892-8), types of vascular architectures (https://doi.org/10.3390/clinpract12060108) and his importance - prognostic and therapeutic (DOI: 10.26355/eurrev_202007_22271)
Add comparison with the study: https://doi.org/10.1515/revneuro-2020-0062
Attention to linguistic nuances: VEGFR overexpression and VEGF levels have no role in the development of gliomas, but are a consequence of the hypoxic and acidosis phenomena present in these entities. Therefore, their levels (factor and receptor) will be increased in high-grade gliomas, where angiogenesis is a characteristic.
Author Response
All of the changes made in the manuscript based on your comments/suggestions are highlighted in yellow color.
Comments 1: The abstract must be structured.
Response 1: Thank you very much for your suggestions for our manuscript. We have structurized the abstract of our manuscript.
Comments 2: VEGFR - is an abbreviation for Vascular Endothelial Growth Factor Receptor, correct this aspect.
Response 2: Thank you very much for your suggestions. We have revised this.
Comments 3: Enter the keywords in alphabetical order
Response 3: Thank you very much for your suggestions. We have made our keywords in alphabetical order.
Comments 4: References are placed in square brackets before the point and at the end of the quoted paragraph, not after each statement.
Response 4: Thank you very much for your suggestions. We have corrected the reference format in our manuscript.
Comments 5: Line 40, please adapt the diagnoses according to the last classification (the same aspect in the discussion chapter).
Response: Thank you very much for your suggestions. However, this statement is not intended to explain the classification of glioma. This statement only explains that according to the research, the survival rates from several types of glioma are relatively poor, even with active treatments. Moreover, the corresponding research only has data for low-grade astrocytoma, degenerative astrocytoma, and glioblastoma multiforme. Therefore, we only have mentioned these types of glioma in the Introduction section.
Comments 6: Line 46: add studies: https://doi.org/10.1038/s41582-019-0220-2 to support reference 6.
Response 6: Thank you very much for your suggestions. We have added that study as reference number 7.
Comments 7: Paragraph 48-63 - can be excluded, it has nothing to do with the subject itself
Response 7: Thank you very much for your suggestions. We have deleted this paragraph.
Comments 8: Lines 69-72 – the statements are repetitive.
Response 8: Thank you very much for your suggestions. We are sorry for the repetitive statements. We have deleted these statements.
Comments 9: Materials and Methods: Given the fact that the study has only two authors, they are probably the same ones who performed the searches and systematized the analysis of the cases. Therefore, reformulate the paragraphs (the idea of ​​several uncredited authors in the article is induced).
Response 9: Thank you very much for your suggestions. We have reformulate the paragraphs to reduce the confusion from the readers regarding the role of each author. We have put the initials from the two authors (P.D.P. and E.J.W.) after each statement that described steps of systematic review and meta-analysis in order to make clear that all steps were performed by the same two authors.
Comments 10: The company and state must be added to the software used in the analysis.
Response 10: Thank you very much for your suggestions. The company name for the software used in the analysis has been mentioned previously. Cochrane Collaborations is the one who developed Review Manager software. We have added the country location of Cochrane Collaborations.
Comments 11: Table 1 - degrees are written in Arabic numerals (review this aspect throughout the text). Add lines to the table, where there are many histological types it is difficult to follow the column they belong to.
Response 11: Thank you very much for your suggestions. We have changed all of the degrees into Arabic numerals. We also have added lines/borders for all of the tables in our manuscript.
Comments 12: For the p value, 3 digits are passed after the point
Response 12: Thank you very much for your suggestions. We have changed the p-value into maximal 3 digits after the point.
Comments 13: Table 3 – find another term for Country (eg. Region?)
Response 13: Thank you very much for your suggestions. We have changed the term "country" into "region".
Comments 14: Start discussions with the other unspecified role of VEGF – angiogenesis (https://doi.org/10.1007/s12035-020-01892-8), types of vascular architectures (https://doi.org/10.3390/clinpract12060108) and his importance - prognostic and therapeutic (DOI: 10.26355/eurrev_202007_22271)
Response 14: Thank you very much for your suggestions. Since the first paragraph of Discussion section from a systematic review and meta-analysis is usually filled with summary of findings, we have provided the discussion about the role of VEGF in glioma, including its ability to serve as prognostic marker and therapeutic target in the second and third paragraph of Discussion section. We also have cited those studies that you have mentioned.
Comments 15: Add comparison with the study: https://doi.org/10.1515/revneuro-2020-0062
Response 15: Thank you very much for your suggestions. We have added the comparison with the study that you mentioned. We also have cited that study.
Comments 16: Attention to linguistic nuances: VEGFR overexpression and VEGF levels have no role in the development of gliomas, but are a consequence of the hypoxic and acidosis phenomena present in these entities. Therefore, their levels (factor and receptor) will be increased in high-grade gliomas, where angiogenesis is a characteristic.
Response 16: Thank you very much for your suggestions. We have revised the statement. We have removed the phrase or statement that implies about the role of VEGF levels and VEGFR overexpression in the development of glioma. Instead, we have stated that VEGF levels and VEGFR overexpression reflect the progression of glioma.
Reviewer 2 Report
Comments and Suggestions for Authors
The authors conducted a systematic review and meta-analysis to explore the association between VEGFR2 gene polymorphism rs2071559 and the risk of developing glioma and aimed to assess whether this specific polymorphism could serve as a biomarker for glioma susceptibility. The article is a well-executed systematic review and meta-analysis that adds to the understanding of glioma susceptibility related to VEGFR2 polymorphisms. It adheres to high academic standards and offers a solid foundation for further studies in this important area of cancer genetics.
While, the use of VEGFR2 rs2071559 as a biomarker for glioma risk has some limitations. Here are a few concerns identified:
1. The association of rs2071559 with glioma risk has been predominantly studied in specific populations, particularly in the Chinese population. There is a lack of comprehensive data across diverse ethnic groups, which limits the generalizability of the findings. Studies focused on a single ethnic group may not yield results that are applicable worldwide.
2. Glioma is a complex disease with multiple contributing factors including genetic, environmental, and lifestyle factors. The rs2071559 polymorphism's role is just one aspect of the genetic predisposition. Gliomas exhibit a high degree of heterogeneity, and thus, relying on a single SNP as a biomarker might oversimplify the pathology of the disease.
3. Although some studies have shown an association between the rs2071559 polymorphism and glioma, the biological mechanisms by which this SNP influences glioma development are not well understood. More research is needed to understand how changes in VEGFR2 function or expression contribute to glioma pathophysiology.
4. In clinical settings, there are already some drugs targeting the VEGFR2 receptor, such as sorafenib. Sorafenib has potential therapeutic effects on a variety of receptor tyrosine kinases (RTKs) associated with tumor growth and angiogenesis, including for the treatment of various cancers such as glioblastoma. In glioblastoma, sorafenib primarily targets receptors such as VEGFR2, PDGFR, and RAF kinases, which are key components of the MAPK signaling pathway. This pathway is crucial for the proliferation and survival of glioblastoma cells. Based on the references cited in this article, it raises the question of whether the polymorphic site rs2071559 in VEGFR2 is related to drug sensitivity and clinical treatment efficacy.
6. The cohorts included in this review are relatively small, which may introduce bias.
7. It is suggested that the article includes explanations of specific concepts like dominant and recessive genetic patterns to cater to a broader audience.
Overall, the English used in this article is well-written. The language is clear and the structure is logical. This contributes to the readability and the effectiveness of the communication within the article.
Author Response
Comments 1: The authors conducted a systematic review and meta-analysis to explore the association between VEGFR2 gene polymorphism rs2071559 and the risk of developing glioma and aimed to assess whether this specific polymorphism could serve as a biomarker for glioma susceptibility. The article is a well-executed systematic review and meta-analysis that adds to the understanding of glioma susceptibility related to VEGFR2 polymorphisms. It adheres to high academic standards and offers a solid foundation for further studies in this important area of cancer genetics.
Response 1: Thank you very much for your valuable comments for our manuscript. We really appreciate your words and suggestions. The changes made on the manuscript based on your comments/suggestions are highlighted in green color.
Comments 2: While, the use of VEGFR2 rs2071559 as a biomarker for glioma risk has some limitations. Here are a few concerns identified: 1. The association of rs2071559 with glioma risk has been predominantly studied in specific populations, particularly in the Chinese population. There is a lack of comprehensive data across diverse ethnic groups, which limits the generalizability of the findings. Studies focused on a single ethnic group may not yield results that are applicable worldwide.
Response 2: Thank you very much for your comments/suggestions. We agree with the Reviewer on this matter. That's why we have already stated this issue as the limitation of our study as follows: "The majority of study's research was exclusively carried out in Asian countries, specifically China. Therefore, the applicability of these results may be limited, especially when considering groups that are not of Asian origin."
Comments 3: Glioma is a complex disease with multiple contributing factors including genetic, environmental, and lifestyle factors. The rs2071559 polymorphism's role is just one aspect of the genetic predisposition. Gliomas exhibit a high degree of heterogeneity, and thus, relying on a single SNP as a biomarker might oversimplify the pathology of the disease.
Response 3: Thank you very much for your comments/suggestions. We totally agree with the Reviewer on this issue. We have changed our statement regarding rs2071559 as biomarker of glioma in the Conclusion section into "Therefore, our study unveils the evidence regarding the involvement of rs2071559 from KDR/VEGFR2 gene polymorphism in the development of glioma, especially among Asian populations." We also have mentioned about exploration of only single SNP in the limitation of our study as follows: "Third, our current research solely investigates the impact of a single SNP, specifically KDR/VEGFR2 rs2071559, on the development of glioma. Our current study did not assess the other SNPs from the KDR/VEGFR2 gene and also other genes that may contribute to the development of glioma. In addition, it is important to bear in mind that environmental and lifestyle factors might also exert an influence on the development of glioma."
Comments 4: Although some studies have shown an association between the rs2071559 polymorphism and glioma, the biological mechanisms by which this SNP influences glioma development are not well understood. More research is needed to understand how changes in VEGFR2 function or expression contribute to glioma pathophysiology.
Response 4: Thank you very much for your suggestions. Actually, we have stated several possible explanations regarding how changes in VEGFR2 expression may influence glioma pathophysiology in the second, third, and fourth paragraph of Discussion section. However, we also agree with the Reviewer that more research is still needed to give better understanding regarding how changes in VEGFR2 function or expression may contribute to the development of glioma. We have mentioned about this as suggestion for future research.
Comments 5: In clinical settings, there are already some drugs targeting the VEGFR2 receptor, such as sorafenib. Sorafenib has potential therapeutic effects on a variety of receptor tyrosine kinases (RTKs) associated with tumor growth and angiogenesis, including for the treatment of various cancers such as glioblastoma. In glioblastoma, sorafenib primarily targets receptors such as VEGFR2, PDGFR, and RAF kinases, which are key components of the MAPK signaling pathway. This pathway is crucial for the proliferation and survival of glioblastoma cells. Based on the references cited in this article, it raises the question of whether the polymorphic site rs2071559 in VEGFR2 is related to drug sensitivity and clinical treatment efficacy.
Response 5: Thank you very much for your suggestions. However, our current research do not focus on the drug sensitivity and clinical treatment efficacy. Our main purpose is to give better understanding about the role of rs2071559 from VEGFR2 gene polymorphism in the development of glioma. This issue can be explored by future research, but not our current research. We have mentioned about this at the end of Discussion section.
Comments 6: The cohorts included in this review are relatively small, which may introduce bias.
Response 6: Thank you very much for your suggestions. Actually, we already have stated about this issue as the limitation of our study. However, based on your suggestions, we have added more statement regarding the potential bias that may arise from small sample size.
Comments 7: It is suggested that the article includes explanations of specific concepts like dominant and recessive genetic patterns to cater to a broader audience.
Response 7: Thank you very much for your suggestions. We really appreciate your suggestions. However, we think that dominant and recessive genetic patterns is common knowledge, especially among healthcare professionals. Since our study is intended to be read by healthcare professionals (e.g. general practitioner, neurologist, neurosurgeon, oncologist, internal medicine doctor, and geneticist), we think that this information is no longer needed to be put specifically in our article. This information can be obtained easily just by doing some search in the Google.
Round 2
Reviewer 1 Report
Comments and Suggestions for Authors
Congratulations!
Reviewer 2 Report
Comments and Suggestions for Authors
Based on the thoroughness of the research, the clarity of writing, and the revisions made in response to reviewers' comments, the article titled "Vascular Endothelial Growth Factor Receptor 2 (VEGFR2) rs2071559 gene polymorphism and the risk of glioma: A systematic review and meta-analysis" is considered suitable for publication. The study provides a comprehensive analysis, clear evidence, and reliable results regarding the association between the rs2071559 polymorphism and increased glioma susceptibility. Thank you for your valuable contribution to glioma research.